# Chito-Oligosaccharide and Propolis Extract of Stingless Bees Reduce the Infection Load of *Nosema ceranae* in *Apis dorsata* (Hymenoptera: Apidae)

**DOI:** 10.3390/jof9010020

**Published:** 2022-12-22

**Authors:** Rujira Ponkit, Sanchai Naree, Rath Pichayangkura, Alexis Beaurepaire, Robert J. Paxton, Christopher L. Mayack, Guntima Suwannapong

**Affiliations:** 1Biological Science Program, Faculty of Science, Burapha University, Chon Buri 20131, Thailand; 2Department of Biochemistry, Faculty of Science, Chulalongkorn University, Bangkok 10330, Thailand; 3Institute of Bee Health, Vetsuisse Faculty, University of Bern, 3003 Bern, Switzerland; 4Institute for Biology, Martin Luther University Halle-Wittenberg, 06120 Halle, Germany; 5Molecular Biology, Genetics, and Bioengineering, Faculty of Engineering and Natural Sciences, Sabanci University, Istanbul 34956, Turkey

**Keywords:** *Apis*, chito-oligosaccharide, *Nosema ceranae*, propolis

## Abstract

*Nosema ceranae* is a microsporidian that infects *Apis* species. Recently, natural compounds have been proposed to control nosemosis and reduce its transmission among honey bees. We investigated how ethanolic extract of *Tetrigona apicalis*’s propolis and chito-oligosaccharide (COS) impact the health of *N. ceranae*-infected *Apis dorsata* workers. *Nosema ceranae* spores were extracted from the guts of *A. florea* workers and fed 10^6^ spores dissolved in 2 µL 50% (*w/v*) sucrose solution to *A. dorsata* individually. These bees were then fed a treatment consisting either of 0% or 50% propolis extracts or 0 ppm to 0.5 ppm COS. We found that propolis and COS significantly increased the number of surviving bees and lowered the infection ratio and spore loads of *N. ceranae*-infected bees 14 days post-infection. Our results suggest that propolis extract and COS could be possible alternative treatments to reduce *N. ceranae* infection in *A. dorsata*. Moreover, *N. ceranae* isolated from *A. florea* can damage the ventricular cells of *A. dorsata*, thereby lowering its survival. Our findings highlight the importance of considering *N. ceranae* infections and using alternative treatments at the community level where other honey bee species can act as a reservoir and readily transmit the pathogen among the honey bee species.

## 1. Introduction

Honey bees play a major role in agricultural pollination services, so their health and colony strength are critical to our well-being [1,2,3,4,5]. *Nosema* disease or nosemosis is one of the most critical parasites behind the most recent worldwide bee health decline [6,7]. There are three classified species of microsporidia: *Nosema apis*, *N. ceranae,* and *N. neumanni*, that can cause nosemosis in honey bees [8,9,10]. They occur worldwide; each species cross-infects the other host [11,12,13]. Although no outward signs or external disease symptoms are typically observed with *Nosema*-infected honey bees, they are known to have energetic stress and digestive disorders resulting in health problems, a shortened lifespan, and decreased survival. At the colony level, declines have been documented that are also associated with a decline in honey stores [7,14,15,16].

There are very few treatments available on the market for nosemosis. One of the most effective products is a non-natural compound, fumagillin [17], which suppresses the growth of *Nosema* spores in honey bees. However, the effectiveness of this treatment is limited because fumagillin is not effective at killing the mature spore stage of the disease [18]. Likewise, fumigation with acetic acid is also used to kill *N. apis* but can only kill the vegetative forms of the parasite [19]. Furthermore, the use of sulfa drugs to control honey bee diseases has also been reported [20], yet antibiotics have become a rising concern for commercial honey bee products because residues from these treatments can contaminate honey bee products, posing risks to increasing antibiotic resistance in humans from honey bee product consumption, which includes honey [21]. Alternative treatments are also needed to aid in combating the potential for resistance to the few treatments currently available on the market. As these treatments do not provide a sustainable solution against nosemosis, natural product alternatives have been explored to control diseases. They tend to be safer for consumption and are more environmentally friendly [20,22]. Developing such a natural product treatment will therefore benefit beekeepers, who may then have new treatment strategies that can be used to control *Nosema* spp.

To date, two previous studies have demonstrated the use of stingless bee propolis to reduce *N. ceranae*-infection in *Apis cerana* [23] and *A. florea* [16], suggesting that it may have a generally positive effect on bee health. Honey bee foragers produce honey bee propolis by collecting resins from various plant species, which has a potent inhibitory effect on microsporidian development [16,23,24,25,26,27]. Feeding stingless bee propolis showed a significantly higher decline in the germination of *N. ceranae* in the ventricular epithelial cells of honey bees. It increased the treated bees’ survival compared to propolis collected from *A. mellifera* [16,23,24,25,26,27]. In parallel, natural products such as chito-oligosaccharides (or COS) are known to promote antimicrobial activity and cell healing [28,29]. These products can stimulate the immune system in the honey bee, *A. mellifera,* and are efficacious in decreasing the degree of infection of bees with *N. apis* [30]. Moreover, COS increases protein contents of the hypopharyngeal glands, and trehalose level of hemolymph in *N. ceranae*-infected *A. dorsata* workers [31].

We examine the effects of propolis extract and COS treatments on *A. dorsata* bees experimentally inoculated with *N. ceranae* that originated from *A. florea* and were then propagated in *A. mellifera* before inoculation. We aimed first to confirm if *Nosema* spores from *A. florea* could infect *A. mellifera* and further infect *A. dorsata*. We hypothesized that propolis extracted from stingless bees and COS could reduce the virulence and energetic stress of *N. ceranae* infection in *A. dorsata*. We then measured the mortality, infection ratio, and spore load of *A. dorsata* to evaluate the potential use of propolis and COS as a way to control *Nosema* development. This honey bee species is particularly important in Thailand, where it serves as the main pollinator for many crops, and in other tropical Asian countries, where it is often a source of honey. However, despite its importance, this honey bee species is suffering from population declines, particularly due to exotic infections such as *N. ceranae* acquired from introduced *A. mellifera* colonies that are used for beekeeping activities, and also with other native honey bee species, in particular *A. florea*, because they share overlapping foraging ranges where it is suspected that horizontal transmission of the pathogen can occur [5].

## 2. Materials and Methods

### 2.1. Chito-Oligosaccharide Solution Preparation

Chitosan oligosaccharide (COS) solution was prepared by hydrolyzing 1% chitosan, MW of 50–100 kDa, in 1% acetic acid, pH 4.5–5.0, with 1 unit of crude chitinase from *Bacillus licheniformis* SK-1 following standard procedures [32]. The reaction was monitored until the average chitosan molecular weight of 5000 to 8000 Da was achieved. Sodium hydroxide solution (5 N) was added until the solution reached a pH of 9–10 in order to stop the reaction and precipitate the COS product. Ethanol was added to make a final concentration of 50% COS, and then the solution was then incubated on ice for 1 h. The precipitate was centrifuged at 5000× *g* for 15 min, then washed with distilled water until the pH of the suspension was around 7 to 8, then this was washed with 100% ethanol, followed by acetone. COS was collected and air-dried at room temperature. The COS molecular weight was determined by gel permeation chromatography.

The 10^4^ ppm stock of COS consisted of 0.25 g of COS (6081 Da) that was dissolved in 5 mL of *A. dorsata* honey (pH = 3.45) and was adjusted to a final volume of 20 mL with 50% sucrose solution (*w/v*). The 10^4^ ppm stock of COS was then diluted with 50% sterile honey solution in water (*v/v*) to make a 10^2^ ppm concentration. Afterward, 0.1, 0.3, 0.5, 1.0, and 1.5 ppm COS concentrations were prepared in the same manner; the pH of the final solution was 3.77 (pH meter, Mettler Toledo Gmbh, Greifensee, Zurich, Switzerland).

### 2.2. Effect of Chito-Oligosaccharide (COS) on the Survival of Apis dorsata Workers

Combs of *A. dorsata* containing capped brood were collected from three healthy colonies in Samut Songkhram Province, Thailand. Colonies were checked to verify the absence of *Nosema* spores following standard procedures [6,33]. The combs were kept in an incubator at 34 ± 2 °C (Memmert IPP 260, Schwabach, Germany) with 50–55% relative humidity (RH) (Barigo-8861, Schwenningen, Germany). The newly emerged bees were *Nosema*-free, to be used in the experiments. Three replicates of 50 one-day-old *A. dorsata* workers per cage were provided daily with two mL of 0 ppm (control) 0.1, 0.3, 0.5, 1.0, and 1.5 ppm of COS defined as CO, 0.1 ppm, 0.3 ppm, 0.5 ppm, 1.0 ppm, and 1.5 ppm, respectively.

All cages of treatments were reared in an incubator at 34 ± 2 °C with 50–55% RH. The mortality was checked daily for 30 days, and any dead bees at the bottom of the cage were removed and recorded.

### 2.3. Propolis Extraction

Stingless bee propolis was collected from three hives of *Tetrigona apicalis* on a farm in Chanthaburi Province, Thailand. Propolis was extracted following the standard method described by Naree et al. (2021) [31]. We defined a yielding crude ethanol extract as a 100% propolis stock solution. For experimental trials, a 50% propolis in water (*v/v*) solution was prepared.

### 2.4. Spore Preparation and Propagation

*Nosema ceranae* spores were extracted from the *Nosema*-infected colony of *A. florea* obtained from the local area of the Chon Buri Province, Thailand. We first propagated the spores by group-feeding *A. mellifera* workers (5 × 10^7^ spores for 50 bees) collected from colonies located at the honey bee research unit of Burapha University, Chon Buri, Thailand. We kept these caged bees at 34 ± 2 °C (Memmert IPP 260, Schwabach, Germany) with 50–55% RH (Barigo-8861, Schwenningen, Germany) for 14 days. Midguts of propagation honey bees were removed and transferred to a 1.5 mL microcentrifuge tube (Eppendorf, Hamburg, Germany) containing 100 µL distilled water. Spores were extracted based on the standard method described by Fries et al. (2013) [33] and Naree et al. (2021) [26]. They were counted under a light microscope (Olympus CX50, Shinjuku, Tokyo, Japan) using a hemocytometer (Hausser Scientific, Horsham, PA, USA) following the procedure described by Cantwell (1970) [34]. Spores were further centrifuged and then resuspended in 50% (*w/v*) sucrose solution to standardize the concentration to 5 × 10^5^ spores per µL. Spores were further identified as *N. ceranae* using Polymerase Chain Reaction (PCR) technique (Appendix A).

### 2.5. Experimental Infection and Nosema Treatments

Three combs containing capped broods were collected from three healthy *A. dorsata* colonies found in Samut Songkhram Province, Thailand. Each colony was checked to verify the absence of *Nosema* spores following standard procedures [6,33]. The combs were kept in an incubator at 34 ± 2 °C (Memmert IPP 260, Schwabach, Germany) with 50–55% RH (Barigo-8861, Schwenningen, Germany). The newly emerged bees were removed and checked to verify the absence of *Nosema* spores following standard procedures [6,33]. *Nosema*-free honey bee ages ranged between 1 and 2 d were caged in groups of 50. These caged bees were divided into four groups (three replicate cages per group). Each bee was fed individually by force-feeding with 50% sucrose solution (*w/v*) containing 10^6^ *N. ceranae* spores per bee in the first two groups. These groups were then fed daily with 2 mL of 0% or 50% stingless bee propolis extract, defined as NO and NO-50P, respectively. Each bee was fed individually by force-feeding with 50% sucrose solution (*w/v*) in the other two groups, defined as the negative controls. One of these groups was given no propolis treatment known as CO, and the last control group was fed with the daily propolis treatment (CO-50P).

In the second experiment, two groups of bees were fed individually by force-feeding with 50% sucrose solution (*w/v*) containing 10^6^
*N. ceranae* spores per bee. These groups were then fed daily with 2 mL of 0 ppm or 0.5 ppm of COS defined as NO and NO-0.5COS, respectively. Each bee was individually provided 50% sucrose solution (*w/v*), by force-feeding, defined as negative control bees (CO). In the last group, termed COS control bees (CO-0.5COS), each bee was fed with the same amount of 50% sucrose solution, but without the *Nosema* spores, and was given daily 2 mL of 0.5 ppm COS. Each cage from each experiment was provided *ad libitum* distilled water, 50% *w/v* sucrose solution, and pollen mixed (30 g of pollen mixed with 8.5 mL of 50% sucrose solution (*w/v*)), throughout the experiment. All treatments were reared in an incubator at 34 ± 2 °C (Memmert IPP 260, Schwabach, Germany) with 50–55% RH (Barigo-8861, Schwenningen, Germany). The treatments were provided with 1 mL doses at a time in two 1.5 mL micro-centrifuge tubes (Eppendorf, Hamburg, Germany) that were provided daily for 30 d.

### 2.6. Infection Ratio

Five bees from each cage at 14 d p.i. were collected, and their midguts were processed for microscopic examination by following the standard protocols described by Suwannapong et al. (2018) [16]. The infection ratio was calculated as the proportion of infected to non-infected cells based on the standard procedure of Higes et al. (2007) [9]. Honey bees from the control group were investigated using the same methods [16].

### 2.7. Infectivity (Spore Load)

Dead bees (collected daily) and all living bees (collected at the end of the experiment, 30 d p.i.) from each group were collected to examine the spore load of each bee. The midguts of all treatment honey bees were removed individually and homogenized in a microcentrifuge tube (Eppendorf, Hamburg, Germany) containing 100 µL of distilled water. Then, the spore load of each honey bee was counted following the standard method described by Cantwell (1970) [34], using a hemocytometer (Neubauer, Portland, OR, USA) under a light microscope (Olympus CX50, Shinjuku, Tokyo, Japan) at 400× magnification.

### 2.8. Statistical Analyses

The data of infection ratios of all treatments were normally distributed (Jarque–Bera JB test: *p* > 0.05), and variances did not differ significantly (Levene’s test: *p* > 0.05). Therefore, we used a parametric one-way ANOVA and Turkey’s multiple comparison test to compare the treatment groups. In addition, the data of the infectivity and percentage of surviving bees of all treatments were normally distributed (Jarque–Bera JB test: *p* > 0.05) but had unequal variances (Levene’s test: *p* < 0.05). Therefore, we used a non-parametric Kruskal–Wallis test and a Mann–Whitney U test to compare the treatment groups. Multiple comparisons were accounted for using a Bonferroni correction. Survival was analyzed using a Kaplan–Meier survival analysis and by plotting the number of surviving bees against days from the initiation of the experiment. Honey bee survival rates were compared across the treatment groups using a non-parametric, univariate analysis of variances and a corresponding post hoc test (Kruskal–Wallis with a post hoc Mann–Whitney U test).

## 3. Results

### 3.1. Effect of COS on the Survival of Apis dorsata Workers

Survival of bees treated with 0.5 ppm COS was significantly higher than that of honey bees treated with 0.1, 0.3, 1.0, and 1.5 ppm COS (*χ*^2^ = 24.67, df = 5, *p* = 0.0001; see Figure 1 and Figure 2). The highest percentage of surviving bees (mean ± SE) at the end of the experiment was found in 0.5 ppm COS group with 77 ± 3.13%, followed by CO, 0.3 ppm COS and 0.1 ppm COS (68.4 ± 3.14%, 61.8 ± 3.21% and 61.2 ± 2.35%, respectively). Moreover, all bees were dead at 30 d in 1.0 ppm COS and 1.5 ppm COS groups (Figure 2).

### 3.2. Infection Rate and Spore Loads of Propagated Honey Bees, Apis mellifera

The infection rate of propagated bees was 100% (n = 100). The spore loads at 14 d p.i. of *A. mellifera* workers infected with *N. ceranae* originated from *A. florea* at a dosage of 10^6^ spores per bee was 83.22 ± 40.42 × 10^6^ spores per bee. Moreover, the spore load range between a hundred bees was 21.15–167.95 × 10^6^ spores per bee (Figure 3).

### 3.3. Infection Ratio

Honey bees inoculated with 10^6^ *N. ceranae* spores per bee showed the highest infection ratio (88.25% ± 1.53), followed by 50% propolis treated bees (NO-50P = 46.53 ± 1.37%), and *N. ceranae* infected bees provided with 0.5 ppm COS (NO-0.5COS = 45.0 ± 0.71%). No infections were found in any control groups. Propolis extract and 0.5 ppm COS significantly reduced the infection ratios in the *N. ceranae*-infected bees (F = 958.50, df = 2, *p* < 0.0001, Figure 4a). No spores were seen in the ventricular cells of the entire midguts of the control group (Figure 5a). In contrast, numerous *Nosema* spores, stained pink with periodic acid Schiff’s reagent (PAS), were found located all over the cytoplasm of ventricular epithelial cells of *N. ceranae*-infected bees on day 14 p.i. (Figure 5b). *Nosema* spores were located at the apical to the bottom, close to the basement membrane, of the cell cytoplasm (Figure 6a). The enlarged and swollen cytoplasm with dispersed microvilli at the tip of the cell had been observed (Figure 6b). Interestingly, some *Nosema* spores were found deposited mainly in the apical part of the ventricular epithelial cell cytoplasm of *N. ceranae*-infected bees treated with propolis extract (NO-50P, Figure 5c and Appendix A). A similar result was found in *N. ceranae*-infected bees treated with NO-0.5COS (Figure 5d and Appendix A).

### 3.4. Infectivity (Spore Load)

No *Nosema* spores were found in control groups throughout the study. The highest spore count was found in *N. ceranae*-infected bees (NO = 90.89 ± 1.89 × 10^6^ spores per bee) followed by *N. ceranae*-infected bees treated with 50% propolis extract (NO-50P = 31.5 ± 2.51 × 10^6^ spores per bee) and with 0.5 ppm COS (NO-0.5COS = 28.44 ± 2.56 × 10^6^ spores per bee), respectively (*χ*^2^ = 166.9, df = 2, *p* < 0.0001, Figure 4b).

### 3.5. Percentage of Surviving Bees

The *N. ceranae*-infected bees treated with 50% propolis extract (NO-50P) and 0.5 ppm COS (NO-0.5COS) showed significantly greater survival compared with the *N. ceranae*-infected bees without treatment (NO) (*χ*^2^ = 51.87, df = 5, *p* < 0.0001, Figure 4c).

The highest percentage of surviving bees was found in control bees treated with 50% propolis extract (CO-50P = 85.9 ± 1.27% survival) followed by control bees treated with 0.5 ppm COS (CO-0.5COS = 82.8 ± 1.24% survival), control bees with no treatment (CO = 77.8 ± 1.74% survival), *N. ceranae*-infected bees treated with 0.5 ppm COS (NO-0.5COS = 37.8 ± 1.51% survival) and with 50% propolis extract (NO-50P = 35.8 ± 1.53% survival), respectively. Moreover, no surviving bees were found in *N. ceranae*-infected bees with no treatment (NO) at the end of the experiment.

## 4. Discussion

*Nosema ceranae* significantly lowers the survival of the giant honey bee, *A. dorsata*. Based on our results, it also has the potential to lower survival in this honey bee species. However, infected bees treated with stingless bee propolis and COS had significantly reduced infection loads and midgut cells, resulting in more prolonged survival than the infected bees that did not receive any treatment. This is important to note because *N. ceranae* has been documented to infect both native and imported species of honey bees in Thailand [35,36].

Food was provided *ad libitum* for all treatments throughout the experiment, so survival differences cannot be attributed to differences in the amount of food available. Therefore, the lowered survival observed in infected bees is most likely due to the bee’s ability to obtain the energy and nutrition required for survival. This notion is supported by previous findings in which infected bees have difficulty in energy acquisition and nutrient absorption even though *ad libitum* food is provided [15,37,38,39]. This corresponds to the results of Suwannapong et al. (2018) [16], who showed the efficacy of propolis extracted from stingless bees to inhibit *N. ceranae*-infection induced from feeding 80,000 spores per bee in the red dwarf honey bee, *A. florea*. In that study, the increased survival was linked to lower *N. ceranae* loads, lower infection ratios, higher trehalose levels in the hemolymph, and larger hypopharyngeal glands.

Stingless bee propolis is known to have antifungal properties, so we suspect there are specific compounds present in the propolis that effectively limit the reproduction of *N. ceranae* [40]. Previously, it has also been shown that propolis treatment in *A. mellifera* results in increased survival as well [25,26,27]. Still, the bees do not prefer food containing propolis as a form of self-medication [25]. Taken together, these studies suggest that propolis can be used as an effective treatment for a range of honey bee species, i.e., as an alternative natural product for the reduction of *N. ceranae* infection loads and infection ratios. However, propolis treatment would have to be administered, but this would be feasible as the bees readily drink sugar water with propolis extract. For commercial honey bees such as *A. cerana* and *A. mellifera*, the propolis extract could be added to sugar water feeders installed into their hives by beekeepers [23,25,26,27]. This appears to be a general effect. Therefore, if necessary, wild bees such as *A. florea* and *A. dorsata* could also be fed with sugar water in artificial flower feeders with propolis extract to improve their health [16,31].

A previous study showed that spores of *N. ceranae* that infect the honey bee, *A. cerana*, had an abnormal structure after exposure to propolis extract, resulting in the interference and inhibition of spore growth and development [23]. Therefore, it is plausible that propolis itself is responsible for the very low spore load observed in bees treated with stingless bee propolis. Although ethanol was used to extract our propolis solution, previous studies have shown that diluted ethanol alone (35% and 49% ethanol) does not effectively reduce *Nosema* spore loads [16,23], suggesting that propolis is indeed the effective substance that is responsible for a reduction in *Nosema* load.

It has been reported that chitosan and oligochitosan (COS) displayed antimicrobial and antifungal activities [41,42]. We found that the medium concentration of COS (0.5 ppm) showed a slightly higher survival than bees without COS (control bees). In contrast, high concentrations of COS (1.0 and 1.5 ppm) reduced survival compared to control bees. The 0.5 ppm COS also reduced the infection ratio between infected cells and non-infected cells of the midgut epithelium. COS may affect *Nosema* development by increasing the permeability of the plasma membrane [43,44] or may enhance bee immunity, thereby reducing the spore load, which would explain the very low spore load found in bees treated with COS. This corresponds to the results of Saltykova et al. (2018) [45], who showed the potential of chitosan to decrease the infection load of *N. apis* in *A. mellifera*.

Moreover, COS may aid in nutrient digestibility and gut functioning, and these gut modifications may be essential for nutrient availability for the proper functioning of the immune system [28]. However, COS’s precise mode of action for treating *N. ceranae* infection is poorly understood. Therefore, it seems crucial to conduct more comprehensive investigations into how COS affects *N. ceranae* viability to reveal its likely capacity as a novel fungicide.

We show *N. ceranae* that originated in *A. florea* can infect *A. mellifera*, be further transmitted to *A. dorsata*, and develop rapidly in both honey bee hosts. The infection loads were consistently high in both hosts. *N. ceranae* can infect *A. mellifera*, *A. cerana*, and *A. florea* with high infection ratios, so there is a need for an alternative treatment to improve honey bee health in general [16,25,26,27,36]. The pathogen develops rapidly in these hosts, and it completes the intracellular life cycle within 3 d p.i. so there do not appear to be any host compatibility issues across the honey bee species [9,36,46]. Moreover, our study suggests that *N. ceranae*-infected foragers, whether *A. florea* or *A. mellifera*, might provide opportunities for cross-infections to *A. dorsata* because these three honey bee species have shared foraging areas, including foraging from the same flora [5]. It has been reported that *N. ceranae* is more prevalent and better adapted to complete its endogenous cycle in warmer climates [47,48]. The tropical climate in Thailand could be essential in enhancing the infection of *N. ceranae* among *Apis* species. It seems clear that *N. ceranae has* become distributed throughout Thailand [35,36].

*Nosema ceranae* spores have been found in the pollen baskets of infected honey bees, which could be shared with other bees in the same colony, providing a mechanism for transmission [6,49,50]. Therefore, honey bee hive products could also be critical sources of infective *N. ceranae* spores that can be transmitted to other colonies that share the foraging area. The spores germinate within the ventricular epithelial cells and insert polar tubes, releasing sporoplasm into ventricular cells, producing more spores. Lowering spore loads can therefore aid in increasing the survival of infected individuals and reducing the potential transmission to other individuals within a colony. Consequently, bee feces containing *Nosema* spores were observed, providing yet another route of infection within a colony [49,51].

The infection of the honey bee gut causes a suite of metabolic changes in the host, such as lower levels of hemolymph trehalose [14,16]. Even though *Nosema*-infected bees do not show apparent external disease symptoms, it has been reported that *Nosema*-infection can cause digestive dysfunction that may cause a shortened life span, resulting in a reduction in a colony population. Consequently, infected colonies have decreased honey production and crop pollination services [46,52]. The lower survival of bees infected with *N. ceranae* is mainly caused by energetic stress [14]. Our results correspond to the study of Valizadeh [53], who showed that chitosan and peptidoglycan significantly reduced honey bee mortality and reduced spore loads. Therefore, perhaps reducing infection ratios and infectivity from propolis or COS treatment is responsible for the increased bee survival we observe in this study. Further investigation is needed to explore the mechanisms responsible for the increased survival we observe here with propolis extract and COS ingestion. Nonetheless, using ethanolic extracted propolis from stingless bees and COS is a new approach that can be used to control nosemosis and enhance honey bee health.

## 5. Conclusions

*Nosema ceranae* spores isolated from midguts of the red dwarf honey bee, *A. florea* workers, can be propagated in *A. mellifera* workers. These spores can further experimentally cross-infect the giant honey bee, *A. dorsata*. The infection increases over time and results in increased honey bee mortality. In addition, our findings reveal that ethanolic extract of stingless bee propolis and COS can cause a reduction in midgut cell spore loads and infection ratios, resulting in increased survivorship in comparison to giant honey bees than that did not receive any treatment. Our results suggest that propolis extract from stingless bees and COS could be alternative treatments to control *Nosema* disease. Different *Apis* honey bee species are known to share flower sources, providing possible routes of horizontal transmission of *Nosema* spores. These treatments can be mixed into food and fed at the colony level to suppress *Nosema* infection, particularly for managed commercial honey bee species where they may serve as a reservoir for *Nosema* spores. In summary, these natural compounds provide new alternative treatments that can potentially improve honey bee health and enhance colony productivity.

## Figures and Tables

**Figure 1 jof-09-00020-f001:**
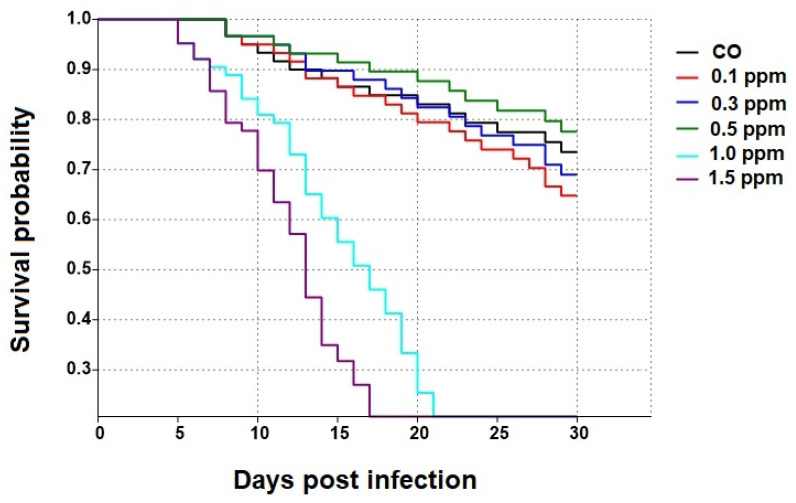
Kaplan–Meier survivorship curves of *A. dorsata* workers provided 0.1, 0.3, 0.5, 1.0, and 1.5 ppm of COS, and 0 ppm (control bees, CO) for 30 days.

**Figure 2 jof-09-00020-f002:**
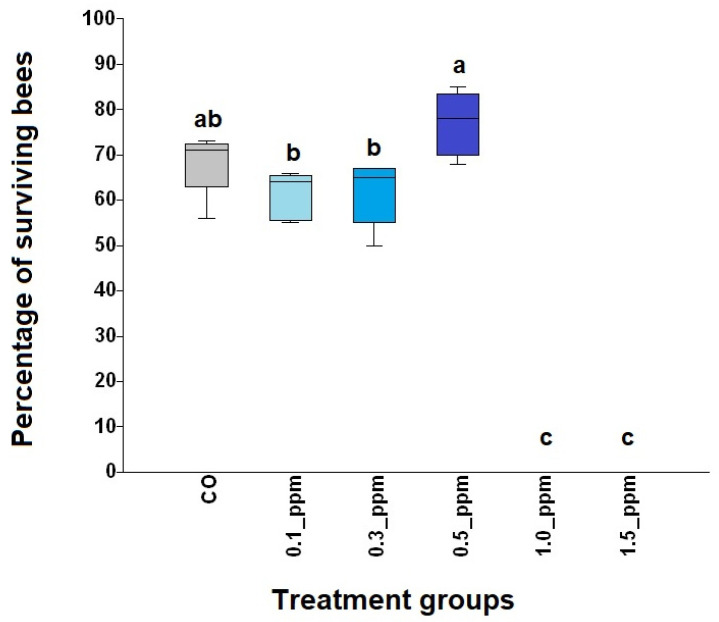
Box plots of the percentage of surviving bees of *A. dorsata* workers provided 0.1, 0.3, 0.5, 1.0, and 1.5 ppm of COS and control bees (CO) for 30 days. Each box represents interquartile ranges from the first to the third quartile, while the line represents the medians, and the whiskers indicate the range. Boxes with different letters above represent significant differences from one to another (Kruskal–Wallis test: *χ*^2^ = 24.67, df = 5, *p* = 0.0001).

**Figure 3 jof-09-00020-f003:**
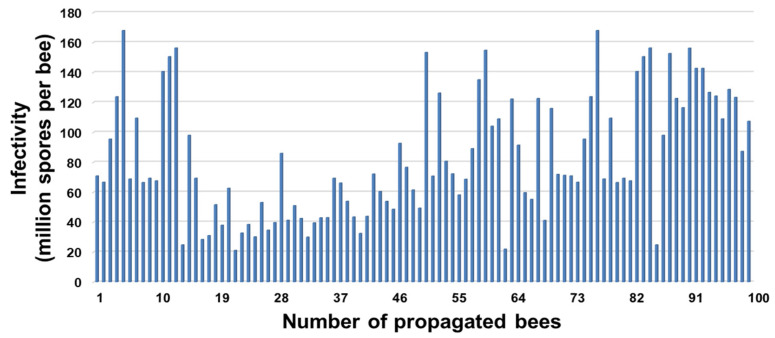
Spore loads (10^6^ spores per bee) of *A. mellifera* workers infected with *N. ceranae* originated from *A. florea* at a dosage of 10^6^ spores per bee for 14 days post-infection.

**Figure 4 jof-09-00020-f004:**
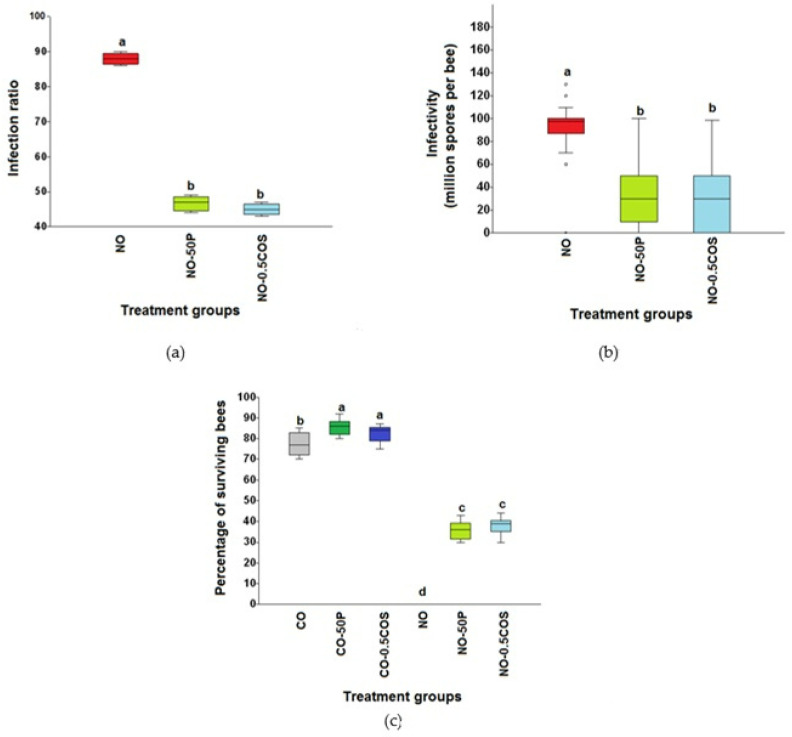
Box plots showed median of infection ratios of *N. ceranae* infected *A. dorsata* bees administered 10^6^ spores per bee on day 14 p.i. (**a**), infectivity (number of spores per bee) from *A. dorsata* after *N. ceranae* infection 14 d p.i. (**b**), and percentage of surviving bees at the end of the experiment after being infected with 10^6^ *N. ceranae* spores (30 d p.i.) (**c**). The CO (gray) was control bees with no treatment, CO-50P (green) was control bees treated with 50% propolis extract; CO-0.5COS (blue) was control bees treated with 0.5 ppm COS. The NO (red) was *N. ceranae*-infected bees with no treatment, NO-50P (light green) was infected bees treated with 50% propolis extract, and NO-0.5COS (light blue) was infected bees treated with 0.5 ppm COS. Boxes with different letters above represent significant differences from one to another (one-way ANOVA: F = 958.50, df = 2, *p* < 0.0001, Kruskal–Wallis test: *χ*^2^ = 166.9, df = 2, *p* < 0.0001, and Kruskal–Wallis test: *χ*^2^ = 51.87, df = 5, *p* < 0.0001 for a, b, and c, respectively).

**Figure 5 jof-09-00020-f005:**
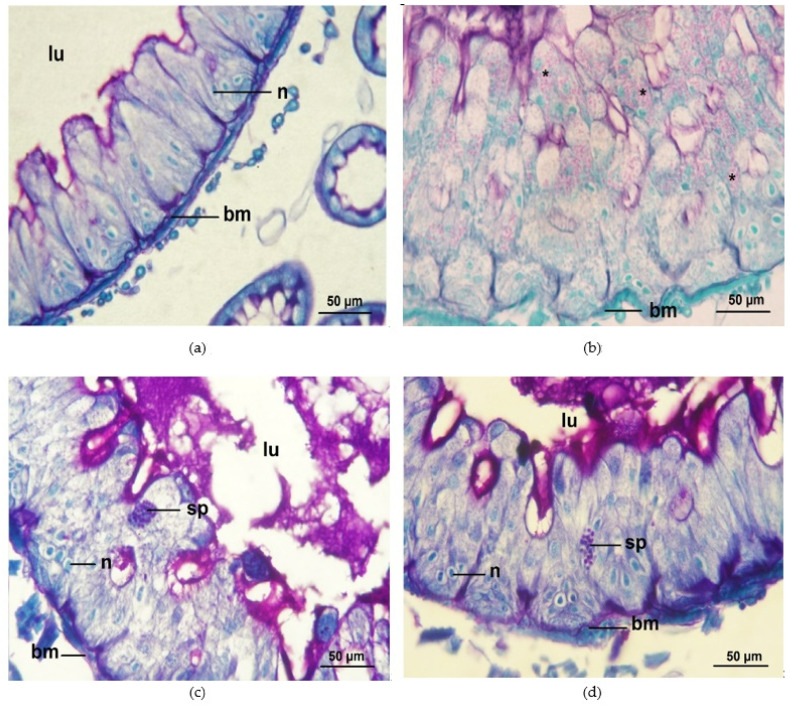
Light micrographs of midgut ventricular cells from *A. dorsata* workers; (**a**) is from a control bee (CO) on day 14 p.i. (PAS, 400x). (**b**) is from a *N. ceranae*-infected bee dosed with 10^6^ spores per bee (NO) on day 14 p.i. (PAS, 400x); (**c**) is from a *N. ceranae*-infected bee treated with 50% propolis extract (NO-50P) on day 14 p.i. (PAS, 400x). (**d**) is from a *N. ceranae*-infected bee treated with 0.5 ppm COS (NO-0.5COS) on day 14 p.i. (PAS, 400x). Abbreviations: bm, basement membrane; lu, lumen of the midgut; n, nucleus of a ventricular cell; sp/*, *N. ceranae* spore.

**Figure 6 jof-09-00020-f006:**
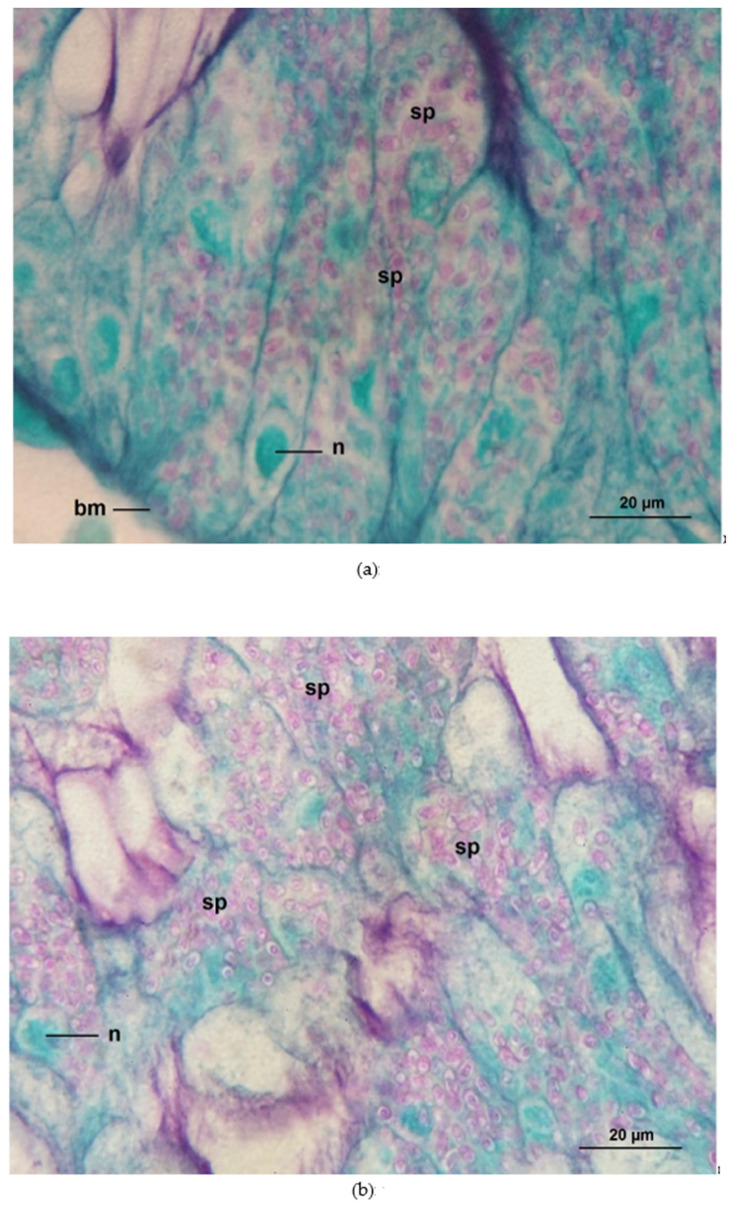
Light micrographs of *A. dorsata* ventricular cells infected with 10^6^ *N. ceranae* spores per bee (NO) on day 14 p.i. (PAS, 1000x). (**a**) *Nosema* spores were located at the apical to the bottom, close to the basement membrane, of the cell cytoplasm; (**b**) the enlarged and swollen cytoplasm with dispersed microvilli at the tip of the cell. *Nosema* spores are stained pink with PAS. Abbreviations: bm, basement membrane; n, nucleus of ventricular cell; sp, *N. ceranae* spores.

## Data Availability

Not applicable.

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
