# Peer review of "Chito-Oligosaccharide and Propolis Extract of Stingless Bees Reduce the Infection Load of Nosema ceranae in Apis dorsata (Hymenoptera: Apidae)"

_jof, 2022, doi:10.3390/jof9010020_

Round 1
Reviewer 1 Report
Dear Editor and Authors,
After reading your manuscript, I realized that you examined Chito-oligosaccharide and Propolis Extract of Stingless Bees Reduce the Infection Load of Nosema ceranae in Apis dorsata (Hymenoptera: Apidae). The issue that you dealt with is well described in the paper, and your work complements previous knowledge and opens up the possibility for further research. The Paper is technically and clearly written. The discussion is clearly written, but this part is very short, please improve them. The methodological details are explained well.
Remarks: The linguistic quality of the paper should be revised more carefully once more by an English expert.
The Introduction does not indicate the status of current knowledge. Moreover, there doesn’t seem to be a clear research hypothesis formulated. Please insert more references about Nosema prevalence in other countries:https://www.sciencedirect.com/science/article/pii/S1319562X19302463
The experimental design is appropriate to resolve the stated objectives of the study. The experimental techniques are appropriate to resolve the stated objectives of the study.
If the suggested changes and information are correctly clarified, I believe that the manuscript is suitable for publication in Journal. Certainly, the results are important to scientific literature. My recommendation is to publish the paper after a MAJOR revision of the manuscript.
Sincerely,
Reviewer 2 Report
The manuscript by Rujira Ponkit et al. describes the application of two natural compounds Chito-oligosaccharide and Propolis in the control of nosemosis among honey bees. The honey bee is one of the important commercial insects due to their crucial role in agriculture. The honey bee serves as the main pollinator for crops, and it is the main source of honey. However, as the natural host of microsporidia, the honey bee is suffering from microsporidia infection which causes the most recent worldwide bee health decline, which will eventually threaten the health of the human being. The experiments that have been done and well-controlled, however, the paper still suffers some minor problems. The Authors should adequately address them if the paper is to be considered for publication in the Journal of Fungi.
Specific comments:
1. The authors identified the ethanolic extracted propolis from stingless bees and COS as new approaches to control nosemosis in the honey bee. However, according to the authors’ introduction and discussion, propolis and COS have been reported to have anti-microsporidia properties, so what is the innovation of this work?
2. What do the letters above the boxes in each BOX PLOTS figures mean? Is that possible to use “*” and p-value to represent the significance between different groups? According to the data in figure 2 and figure 8, there was no significant difference between the treated group and the control, please explain.
3. As Figures 4, 8, and 9 all described similar results caused by the treatment with NO-50P and NO-0.5COS. I would suggest combining those figures into one figure to make the logic of the article more rigorous.
4. Line 236, “No infections were found in all control groups”, what are the control groups? According to the description of the author, the control group is already set as the infected honey bee without treatment with propolis or COS. Were there other controls set here? Please clarify.
5. In Fig. 5b, the magnification is too low and it is not easy to find the spores (sp) in the cell cytoplasm, I would suggest indicating the sp with arrows or stars instead of “sp” in the picture. Besides, figure 5c and figure 7a, figure 5d, and figure 7 b are describing the same phenomenon of the infected tissues from the treatment of propolis and COS. I would suggest moving Figure 7 to the supplementary data.
6. Line 250-256, the author indicated that the spores in the untreated group were located at the apical to the bottom of the ventricular epithelial cell cytoplasm, but the groups treated with propolis or COS were located mainly in the apical part. Is the author describing the infecting location inside the individual epithelial cell cytoplasm? Or the location of infected cells in the midgut? Furthermore, could the author hypothesize the possible reason that causes the changing of infection locations? And does this phenomenon have any relationship with the change in motility, infection, and survival rate caused by the compound's treatment?
7. In figure 6b, please label the enlarged and swollen cytoplasm and dispersed microvilli in the picture.
Round 2
Reviewer 1 Report
Respected,
This paper can be accepted for publication!
All the best!